# Wnt/β-catenin Signaling in Tissue Self-Organization

**DOI:** 10.3390/genes11080939

**Published:** 2020-08-14

**Authors:** Kelvin W. Pond, Konstantin Doubrovinski, Curtis A. Thorne

**Affiliations:** 1Department of Cellular and Molecular Medicine, The University of Arizona, Tucson, AZ 85719, USA; curtisthorne@email.arizona.edu; 2Green Center for Systems Biology, University of Texas Southwestern Medical Center, Dallas, TX 75390, USA; Konstantin.Doubrovinski@utsouthwestern.edu

**Keywords:** Wnt, β-catenin, tissue patterning, tissue homeostasis, tissue organization, self-organization, reaction-diffusion, morphogens

## Abstract

Across metazoans, animal body structures and tissues exist in robust patterns that arise seemingly out of stochasticity of a few early cells in the embryo. These patterns ensure proper tissue form and function during early embryogenesis, development, homeostasis, and regeneration. Fundamental questions are how these patterns are generated and maintained during tissue homeostasis and regeneration. Though fascinating scientists for generations, these ideas remain poorly understood. Today, it is apparent that the Wnt/β-catenin pathway plays a central role in tissue patterning. Wnt proteins are small diffusible morphogens which are essential for cell type specification and patterning of tissues. In this review, we highlight several mechanisms described where the spatial properties of Wnt/β-catenin signaling are controlled, allowing them to work in combination with other diffusible molecules to control tissue patterning. We discuss examples of this self-patterning behavior during development and adult tissues’ maintenance. The combination of new physiological culture systems, mathematical approaches, and synthetic biology will continue to fuel discoveries about how tissues are patterned. These insights are critical for understanding the intricate interplay of core patterning signals and how they become disrupted in disease.

## 1. The Wnt/β-catenin Pathway

The Wnt pathway is a highly conserved cell-to-cell signaling pathway, appearing in metazoans from sponges [1] to nematodes [2] to chordates [3]. Wnt signaling plays essential roles in development, maintenance of tissue homeostasis, and regeneration of damaged tissue. After its role in *Drosophila* patterning and mouse tumor formation was discovered 40 years ago [4,5], Wnt has been extensively studied in virtually all model biological systems and continues to provide both basic biological insights and clues relevant to animal form, function, and disease.

In this review, we will focus on the canonical Wnt/β-catenin pathway as a tissue organizing and self-organizing factor during embryogenesis/development and regeneration/homeostasis. For more general reviews on patterning and self-organization, the reader should also consider References [6,7,8]. β-catenin-independent Wnt pathways also exist, yet their activity has been less extensively studied in the context of tissue organization and patterning. For reviews on β-catenin-independent Wnt signaling, we refer the reader to References [9,10,11,12].

In the canonical Wnt/β-catenin pathway (Figure 1), cells not receiving extracellular Wnt ligands maintain an active cytosolic destruction complex, which serves to degrade the continually expressed transcriptional co-activator β-catenin. The core components of the destruction complex consist of Axin, Adenomatous polyposis coli (APC), casein kinase 1 α (CK1α), glycogen synthase kinase 3 (GSK3), protein phosphatase 2A (PP2a), and the ubiquitin ligase βTrCP. Axin and APC bind β-catenin, which is phosphorylated by CK1α and GSK3, exposing the βTrCP ubiquitin target site on β-catenin [13] and leading to proteasomal degradation. This maintains low cytosolic levels of β-catenin in the absence of a Wnt ligand. Upon binding of extracellular Wnt to the membrane receptors Frizzled and low-density lipoprotein receptor-related protein 6 (Lrp6), Dishevelled is recruited and promotes inhibition and possibly disassembly of APC, Axin, and GSK3. With the destruction complex function suppressed, β-catenin accumulates and translocates to the nucleus to promote expression of Wnt target genes. β-catenin target genes are transcribed through cooperation with several other nuclear factors [14], including the T-cell factor (TCF) transcription factor, which switches roles from a repressor to an activator (reviewed in Reference [15]). This transcription program initiates a downstream cascade of events that can drive proliferation, differentiation, and renewal of the stem cells [16], among other outcomes [17].

## 2. Tissue Self-Organization

The ability of a single cell to give rise to the complexity of a multicellular organism has fascinated scientists for generations. In 1892, Hans Dreisch separated sea urchin embryos at the 4-cell stage and discovered that all separated cells were able to differentiate into complete larvae (Die Biologie als selbstständige Wissenschaft (1893)). This raised the question, what keeps the intact 4-cell embryo from giving rise to four complete larvae? Dreisch’s discovery was critical as it showed that a cell’s fate depends on the external (in this case inhibitory) signaling from proximity or contact with its neighbors. This was also a keystone discovery because it showed that early cells are autonomous units that contain all the information needed to self-organize themselves into complex functional tissues.

Tissue self-organization is a key aspect of animal development and adult tissue homeostasis. We define self-organization as the process by which tissues can form patterns from an initial symmetrical or chaotic state. Self-organization can be driven by diffusible molecules, cell–cell or cell–substrate signaling, and mechanical forces. These triggers allow cells to “sense” their neighbors and environment to adjust proliferation, growth, differentiation, death, etc., accordingly, to insure robust and uniform development, regeneration, and homeostasis. Cell adhesion-based signaling is also essential to tissue organization. For example, the mechanosensing transcription factor YAP is required for the initial symmetry breaking events seen in organoids [18] and crypt formation is also driven by the activity of integrins during gut development [19]. For excellent reviews on adhesion and tissue organization, we refer the reader to References [20,21,22,23].

Tissue self-organizing events can be separated into two categories: embryogenesis/body development and homeostasis/regeneration (Figure 2). Development of the embryo, body axis formation, and appendage development are thought to be unidirectional and temporally regulated patterning processes. Unlike developmental processes, regeneration and homeostasis are meant to maintain a steady state of form and function and must dynamically sense the need to renew or repair tissues during an organism’s adult lifetime. Development of the embryo is the earliest example of tissue patterning, as the symmetrical mass of cells gives rise to high-fidelity organization. Embryology, in addition to being the foundational work in understanding how patterning occurs, continues to provide insights into the mechanisms that govern tissue architecture. Embryogenesis and body development entail organizing events that are initiated by spatial and temporal cues such as compartmentalized germ layer signaling interactions [24] or the depletion of a finite pool of maternally provided mRNAs [25], among many other examples. Tissue regeneration and homeostasis are distinct from embryogenesis and body development as these systems exist throughout the adult life of an organism and allow for tissue-autonomous plasticity and response to tissue damage. Evidence for this comes from the ability of adult tissues to be disassembled or damaged, only to faithfully give rise to their original complex patterns, such as during the re-patterning of culturing of human organoids [26] or the regeneration of severed *Planarians* [27].

A clear example of adult tissue self-organization is the architecture of the mammalian intestinal crypts. Wnt signaling is critical for the establishment and maintenance of the regular periodic pattern of crypt invaginations lining the intestine. Crucially, this periodic pattern arises presumably from interactions of intestinal stem cells and their differentiated progeny [28]. This was demonstrated in recent in vitro reconstitution experiments involving murine small intestine organoids [29]. There, small intestine-derived tissue fragments were cultured to produce a flat two-dimensional (2D) lawn of cells lining the bottom of a culture dish. Strikingly, the cellular lawn comprised regular patterns of circular patches of stem and proliferative cells, thus closely recapitulating the regularity and pattern of crypt structures of those in vivo. These experiments demonstrate that the tissue-level patterns can appear in absence of external spatial cues and when restricted to 2D space. Therefore, there is something intrinsic to the gut epithelial monolayer that drives self-organization.

## 3. Pattern Formation Reaction-Diffusion Models

How could patterns such as the repetitive intestinal crypts be autonomous to the crypt itself? To begin conceptualizing underlying mechanisms, we look back at pattern theory. How patterns in nature can arise from disorder has interested scientists for a long time. Famed mathematician and father of the computer, Alan Turing, was curious himself about the mechanisms underlying patterns found in nature. In 1952, in his only published work dealing with biological systems, he modeled how diffusible molecules (where he first coined the term “morphogen”) could lead to development of pattern formation from an initial random state. Notably, Turing showed through mathematical modeling that stationary polka dots, stripes, and wave patterns could be obtained from two molecules, an activator and its inhibitor, diffusing at different rates [30]. Similarly, Meinhardt and Gierer [31] using non-linear reaction terms and computer modeling discovered that a self-promoting short-range activator coupled with a long-range inhibitor is capable of stable patterning (Figure 3, Left). This mathematical mechanism of morphogen feedback is generally referred to as a reaction-diffusion (RD) model. Although an attractive explanation for patterning, diffusible activator/inhibitor pairs which control biological patterning by an RD mechanism have been challenging to validate in experimental animal models. There is a general concern that two-component RD models are likely over-simplified and cannot explain complex phenotypic variables seen in model systems [7]. Meinhardt and Gierer’s model also relies on large differences in diffusivity between the activator and inhibitor, leading to skepticism about physiological relevance. However, recent work has uncovered many mechanisms by which cells can actively control the diffusivity of a morphogen (discussed below). Additionally, contemporary RD models have shown that the diffusivity of the system can be overcome through the addition of a third morphogen node, allowing a bistable pattern to occur even if all morphogens have equivalent diffusivity, if they are tightly controlled by multiple feedback loops [32]. However, the principle described by Meinhardt and Gierer laid the groundwork for more complex mathematical models that take into account multiple feedback loops and components [33]. Regardless of the number of components and feedback pathways, many examples currently support RD systems in development and maintenance of tissue patterning (Figure 3, Right, and discussed below), of which Wnt signaling seems to be a key player.

RD models rely on critical diffusion properties of morphogens. In the past decade, many mechanisms have been discovered by which cells and tissues can propagate morphogen signals by means other than simple free diffusible molecules and Brownian motion. These other mechanisms include transport by motor proteins [34], mechanical release of tethered molecules [35], morphogen sequestration/separation [36,37], and handoff by unique cellular structures such as telocytes [38] or cytonemes [39,40,41]. In contrast, activators may be restricted to short distances by chemical modifications or physical boundaries, and this process is known as active diffusion. Compelling experiments using laser ablation of striped patterns in zebrafish show a striking resemblance to computer-driven simulations of RD models [42]. Further research on zebrafish patterning has revealed that the different pigment cells physically recapitulate a RD system via short-range mutual inhibition, and long-range activation, which is dependent on Notch [43]. Recent investigations have filled in some of the underlying cellular behaviors that drive the patterning, including an intricate balance of cell migration, proliferation, and death [44]. Murine and other digit developmental patterning is regulated by a RD mechanism, involving Wnt, bone morphogenic protein 2 (Bmp2), and SRY-Box Transcription Factor 9 (Sox9) [45]. Turing recognized that active transport and mechanical forces within a cell or tissue could also play vital roles in the formation of patterns and acknowledged that his model left out other potential mechanisms for patterning. As new technologies develop, we are seeing various ways that tissues can robustly produce autonomous patterns which contribute to development, homeostasis, and regeneration. Zebrafish stripe pattern and murine limb development provide two examples where ostensibly, morphogens interact via short-range activation and long-range inhibition to mediate autonomous pattern formation.

## 4. Length-Scales of RD Models

Across every multicellular organism, self-organization is present at strikingly different scales, from the subcellular mitotic spindle [49] to entire limb development [45]. At first contemplation, it is not obvious at what length-scales RD systems could operate. Are these models only relevant in the low micrometer scale of small clusters of cells or could they pattern larger high micrometer or millimeter tissue space? RD models have been shown to explain signaling of morphogens across a range of distances, from ~5 uM during spindle assembly [49,50] to over 200 uM during zebrafish embryo development [51]. Paracrine signaling between zebrafish pigment cells is essential for RD-like pattern formation after pattern ablation [42], suggesting that RD-like models can explain certain tissue patterning at multicellular scales. In a two-morphogen system, the scale of the RD system is essential in predicting pattern formation as the original models could result in highly variable patterns if spatial parameters are changed [52]. RD models with a smaller diffusion space predict less complicated patterns such as stripes and larger space predicts spots or labyrinth patterns [52]. This led scientists to speculate that this is the reason that most patterned mammals have spotted coats and striped tails rather than vice versa. At the cellular level, the diffusion constant, comprised of the ability of the morphogen to move and degrade, limits the distance at which a RD model can exist. Wnt-driven patterning predicted by a RD model has been shown at the level of whole appendages, and perturbation of Wnt or BMP results in expected patterning defects, such as reduction or absence of digits [45]. In most model systems, it is still unclear the length-scales of RD morphogens in vivo. This is because in order to fully test an RD model, one must be able to tune the reaction or diffusion properties of the morphogen while maintaining its biological activity. In *Drosophila*, limiting the morphogen diffusion by tethering has been used to uncouple the roles of short- and long-range morphogens such as Wnt [53], and suggests that Wnt can act both in a long- and short-range capacity to achieve correct pattern formation, described in detail below.

## 5. The Wnt Morphogen

A modern definition of a morphogen is a diffusible substance whose non-uniform distribution instructs tissue pattern formation and cell type specification. Wnts are classically defined as morphogens due to their clear function in tissue patterning and cell fate. Key components of the Wnt pathway were first identified in *Drosophila* embryo segmentation mutants [4,54], first demonstrating the role of the Wnt pathway in patterning the embryo and possible symmetry breakage. Additionally, pioneering work from *Xenopus laevis* embryos showed that ectopic expression of Wnt1 mRNA was sufficient to induce the formation of a second body axis [55]. This suggested that Wnt signaling can instruct the cell fates of whole tissues during development. As mentioned above, morphogen diffusion is a key component of pattern formation from spontaneous self-organization, and in the Meinhardt and Gierer RD model, a short-range activator is essential [31]. Many contemporary RD models are built using the knowledge that ligands can diffuse via active diffusion [56] as the mechanism controlling a morphogen’s range. Multiple mechanisms for active diffusion of Wnt have been described, allowing tissues to tightly regulate the distance at which Wnts can act.

### 5.1. Short-Range Wnt

A highly conserved post-translational lipid modification that decreases its ability to diffuse long distances is essential for proper Wnt secretion. In order to leave the endoplasmic reticulum for eventual secretion, the Wnt protein must be mono-palmitoylated by the *O*-acetyltransferase porcupine (porcn) [57]. Loss of the PORCN gene results in gastrulation failure [58] and abnormal murine uterus development [59]. In early work, the *Drosophila* homolog of human Wnt, Wingless (Wg), was found within vesicle-like structures as well as in the spaces between cells, suggesting that that molecule acts in a paracrine signaling capacity [60]. Since then, secretion of Wnt within exosomes that restricts Wnt ligand action to shorter distances has become well established [61]. Other short-range mechanisms have been shown to promote the effective propagation of Wnt signaling across cells from a secretory hub. Cell membrane carrier proteins such as heparan sulphate proteoglycans are essential for Wnt-dependent *Xenopus* axis formation [62]. Wnt can be handed off directly by contact with secretory cells in the gut to enable stem cell renewal [63]. Wnt can also be directly delivered to cells via specialized filopodia carrying payloads of Wnt to cells harboring Frizzled receptors at a distance [40,41]. In organoid systems, co-culture of murine small intestine organoids with Wnt knockout organoids does not rescue the knockout phenotype, and direct contact with Wnt secreting Paneth cells is required to rescue knockout cells, suggesting that Wnt acts at short distances to maintain the homeostasis of the gut [63]. The control of Wnt over cell fate has been directly observed in cells as daughter cell fate can be pre-determined by asymmetrically triggering through addition of a Wnt-coated bead on one side of the cell [64]. These data highlight the importance of short-range secretory Wnt during morphogenesis.

### 5.2. Long-Range Wnt

Wg, the Wnt homolog in *Drosophila*, acts across large distances in the development of the wing [65], which relies on lipid binding cofactors, such as secreted Wg-interacting molecule (Swim) [66]. Wg immobilized by tethering to the cell membrane has been a useful tool to study the role of long-range Wnt signaling. Original results showed only mild defects in wing size and normal external morphology in Wnt-immobilized wings [53], suggesting Wg secretion is dispensable for long-range signaling. However, a closer look at the effects of membrane-tethered Wg has revealed significant phenotypes at the tissue level. In particular, defects were described in stem cell renewal and epithelial fate determination [67,68]. During murine limb development, Wnt-driven patterning predicted by an RD model has been shown at the level of whole appendages, and perturbation of morphogens results in expected patterning defects, suggesting action at a distance [45]. However, it is unclear the distance at which a single Wnt molecule acts in this system, as Wnt proteins were not monitored directly. Whatever the methods secretory cells use for the transfer of Wnt ligands, one can envision a chemical gradient occurring as cells are positioned further from Wnt-secreting cells.

## 6. Self-Organization during Embryogenesis

Studies of the embryo provide an ideal setting for examining the patterns which arise from stochasticity. The term “symmetry breaking” is commonly used to describe the first signs that tissue architecture is beginning to form, as seen during the formation of the intestinal crypt from a planar sheet of epithelium [19]. However, from a morphogen perspective, symmetry breaking also marks the key event towards a steady-state pattern essential in developing and adult tissues (Figure 3, left). Arguably, the most influential experiment in embryology was conducted by Hilde Mangold in 1924. She showed that cells from one salamander blastopore could be grafted onto another embryo to instruct the surrounding tissue and direct cell fate [69]. This transplant of cells became known as Spemann’s organizer, after Mangold’s advisor, Hans Spemann. The difference in pigment from the two different salamander species used allowed for tracking of cells as they organized and developed into a secondary body axis harboring a functional nervous system derived from the transplanted clones. This work elegantly demonstrated that an unknown factor or factors intrinsic to the originating tissue provided sufficient information to reprogram other cells and produce a secondary body axis. An essential factor for the development of the resulting secondary body axis in Mangold’s famous experiment was a member of the Wnt gene family [70,71]. Following Mangold’s experiment, morphogens that interact with Wnt signaling were discovered, such as bone morphogenic proteins (BMPs), that work in concert with Wnt ligands to balance differentiation and stemness and regulate tissue patterning in *Drosophila* [72], Xenopus [73], and murine models [29,47,74].

In the developing embryo, Wnts are stemness-promoting activators. Other morphogens, such as BMPs, can act upon Wnts to establish the embryonic tissue patterning. In human embryos, BMP triggers a Wnt/Nodal positive feedback loop that initiates the development of the early germ layers [75]. In Xenopus, the morphogen Lefty inhibits Wnt, which is essential for normal patterning and gastrulation [76]. Experiments using fluorescence recovery after photobleaching (FRAP) in developing zebrafish have described the relationship between the diffusible morphogens Nodal and Lefty, which are essential for axis formation [77], to regulate each other in a RD-like manner [78]. Other morphogens such as sonic hedgehog and transforming growth factor β (to name a few of many) are also essential for proper patterning during morphogenesis, reviews for which can be found in References [79,80].

## 7. Self-Organization during Development

### 7.1. Digit Patterning

The developing limb manages to accomplish the regular spacing and size of digits faithfully in many mammals. The bones of the hand show digitation well before the formation of fingers: what then controls the spatial patterning of the bones? This common yet remarkable three-dimensional (3D) patterning has been an active area of study to discover the critical regulators of tissue patterning during development. Using murine digit development as a patterning model, Raspopovic et al. [45] made an important discovery by combining math models and murine developmental models. The authors discovered the components (Wnt, Bmp2, and Sox9) responsible for digit patterning and confirmed their relationship was indeed a RD system of regulation where Wnt and BMP promote and oppose Sox9 respectively, to establish precise tissue patterning [45]. Subsequently, this RD system has been shown to be conserved in the digit patterning of pectoral fins and development of limb digits involve spot patterning followed by stripe formation both predicted by mathematical RD models [48,81]. These data provide strong evidence that a RD mechanism involving Wnt underlies the faithful development of digitation patterns in animals.

### 7.2. Hair Follicle Patterning

The regular spacing of mouse hair follicles (Figure 3, right [82]) are a useful tool for the study of tissue patterning and the role of morphogen gradients. During development, arrays of evenly spaced hair placodes emerge and organize themselves after being exposed to a Wnt morphogen gradient [83]. This Wnt gradient is opposed by surrounding BMP that limits the size and distribution of hair placodes [84], a proposed RD system similar to that seen in the patterning of proliferative regions of 2D intestinal organoids [29]. Upon activation of the Wnt pathway, the Wnt inhibitor, Dickkopf (Dkk), is expressed and downregulates Wnt signaling via its interaction with the Wnt co-receptor Lrp6 and the membrane receptor Kremen [85,86]. This Wnt/Dkk negative feedback was shown to regulate hair follicle spacing and interrogation of this system produced the predicted phenotypes generated from a RD model [87]. Together, these data suggest that hair follicle spacing is consistent with a RD system involving Wnt and at least two inhibitor pairs.

## 8. Self-Organization during Tissue Homeostasis

Self-organization has been studied in detail in the context of development, where progenitor cells are instructed to differentiate into appropriate tissue layers and organs to serve a specific function during the life of an organism. However, self-organization is not limited to embryo development and is especially important in maintenance of adult tissues. Most tissue in the human body is in a constant state of self-renewal. Tissue such as the lining of the gut, skin, muscles, and even bone, are constantly replacing damaged cells, though the renewal rates vary wildly among various tissues. Wnt/β-catenin signaling is necessary and sufficient to instruct tissues to transition to a self-renewing state. Introduction of purified Wnt3a induces hematopoietic stem cell renewal, supporting its role in somatic stem cell renewal [88]. Further, addition of excess Wnt or inhibition of the β-catenin destruction complex results in the abolishment of differentiation in intestinal organoid cultures, showing the importance of balancing Wnt signaling for tissue homeostasis [89]. Many other examples exist which show how critical Wnt signaling is in the maintenance of the adult stem cells (reviewed in Reference [16]).

Organoids have proved a useful tool to reveal the existence of tissue autonomous organizing behavior. By definition, organoids form organ-like structures ex vivo when removed from surrounding tissues. There are currently organoid methods available for almost every major organ in the body [90]. The gut is in a constant state of renewal as progenitors originate in the crypts, move out to divide, and eventually differentiate into one of many cell fates. The high turnover of this tissue compared to others makes the gut an ideal place to study the maintenance of tissue homeostasis. The gut also displays remarkable patterning: when looking at a cross-section of the intestine (Figure 3, right), one immediately sees the repetitive glandular pattern made by the crypt structure and the resident stem cell niche inhabited by the adult intestinal stem cells [91].

## 9. Self-Organization during Regeneration

When normal tissues are damaged and a niche (node) is lost, a self-organizing system will renew and repattern the damaged tissue. This plasticity provides a dynamic but controlled environment where cell fates are induced depending on need. Tissue level self-organization is critical in the process of regeneration where damaged tissue must reorganize itself robustly to reform functional tissue. Below, we highlight a few examples of the Wnt-driven tissue regeneration.

### 9.1. Hydra and Planarian Regeneration

An early example of Wnt-dependent regeneration pathways comes from Cnidarian model systems. *Hydra* has been an exceptional model to study regeneration of tissues as the head can be removed to induce regeneration [92]. This process begins within 30 min, and a differentiated head structure can be observed within 30 h. In *Hydra,* during tissue regeneration, β-catenin and TCF transcription is upregulated within 30 min and Wnt protein expression is localized to the regenerating tips within an hour, and regeneration-deficient mutants lack Wnt signaling [93]. In *Planarians* (another useful model system due to their striking regeneration ability), the Wnt pathway antagonizes anterior development, as β-catenin knockdown animals which are damaged form multiple head structures at the regenerative sites, while APC knockdown results in multiple tails during regeneration [94].

### 9.2. Zebrafish Neuromasts

Regularly patterned skin cells provide a useful tool for studying self-organization during development as well as during regeneration, as the patterning can be easily visualized in 2D. Zebrafish harbor symmetrically patterned sensory organs call neuromasts which are continuously renewed and can regenerate after damage. As in hair follicle placode formation [95], Wnt is also essential for the formation of zebrafish neuromasts [96]. Both budding of new neuromasts and regeneration of neuromasts by neomycin ablation was dependent on a Wnt/Dkk regulatory network, where Wnt-producing cells promote cell proliferation and development of hair cells, which then secrete Dkk to limit the size of both developing and regenerating neuromasts [97]. These data suggest yet another 2D patterning regulated by a potential RD activator/inhibitor pair involving the Wnt morphogen.

### 9.3. Human and Mouse Regeneration

Adult tissues will sustain some form of damage during an organism’s life, and therefore must maintain the ability to regenerate and re-organize lost tissue. The human gut is not only short lived but also has a remarkable ability to regenerate and self-organize after acute damage. Certain tissue cells types become short lived as they differentiate and serve their purpose, such as the absorptive and secretory cells within the gut. A robust self-organizing system exists within the gut as it can not only renew rapidly, but also can sense when specific cell types are depleted and replace them, such as during damage and regeneration. In humans and mice, regeneration of damaged tissue is dependent on canonical Wnt signaling, as Wnt drives the renewal of Lgr5+ stem cells which renew damaged cells in the intestine [98]. A striking example of gut regeneration comes from its ability to quickly renew eradicated niches. Normal small intestine can induce de-differentiation and replenish the entire stem cell niche within a few days of eradication of all stem cells [99]. Also, loss of Wnt-secreting Paneth cells can also be rescued by plasticity in the normal gut, as enteroendocrine and tuft cells will switch roles upon acute depletion of the Paneth cell niche [100]. These data demonstrate the ability and the need of autonomous regeneration mechanisms, such as RD morphogen pairs, to renew and regenerate tissues properly. Murine model systems are commonly used to study tissue regeneration in the gut after acute damage induced by chemical agents such as Dextran Sulfate Sodium (DSS), which is designed to mimic naturally occurring damage associated with inflammatory bowel disease [101]. After DSS induces damage in the gut, the mechanosensing transcription factor YAP1 forms a complex with β-catenin and TCF within the nucleus to allow for proper regeneration and stem cell renewal [102], highlighting the essential role of the Wnt/β-catenin during regeneration. Other chemical models of tissue damage are also used to study specific gut damage events and regeneration is dependent on Wnt in some cases. For example, loss of Wnt signaling potentiates acute intestinal damage induced by LPS, hypoxia, and hyperosmolar solution, used to simulate a severe disorder known as Necrotizing enterocolitis (NEC). This pathology is due to an imbalance between proliferation and differentiation maintained by Wnt signaling, and administration of exogenous Wnt7b is sufficient to rescue pathological phenotypes [103]. These are just a recent few of many examples demonstrating the critical role of Wnt/β-catenin in regeneration of human and mouse tissues, for more reviews of Wnt signaling in regeneration, please see References [16,104,105,106].

## 10. Cancer and Tissue Disorganization

Self-organization pathways are often dysregulated or hijacked during diseases of aberrant cellular growth such as cancers. Mutations in the Wnt pathway have long been associated with the disorganization seen during tumor progression. The first tool used by pathologists to diagnose cancer is the presence of dysplasia revealed as a lack of proper tissue patterning or superfluous cell growth. In many colon cancers, driver mutations are acquired sequentially, and common initial mutations driving dysplasia are nonsense mutations in APC, a negative regulator of the Wnt pathway [107]. Unchecked Wnt signaling is associated with tumorigenesis [108], and tumor cell growth can be arrested by inhibition of Wnt secretion or stabilization of Axin, both of which will block Wnt signaling through blocking ligand signaling or enhancing β-catenin destruction complex activity, respectively [104,109]. It has been well established that many cancers (~85% in colorectal) are associated with activating mutations in the Wnt pathway [17]. Most of these mutations result in deactivation of the tumor suppressor APC. Whether it be germline deletions resulting in familial adenomatous polyposis, or sporadic mutations identified in colon cancer patients, an overwhelming number of these pathological mutations result in APC truncations, which remove the β-catenin and Axin 1 and 2 (SAMP) binding sites. Mice with truncation mutations in APC which preserve the SAMP regions do not develop tumors [110], whereas truncations removing the SAMP sites cause adenomatous polyposis [111], highlighting the importance of the binding of APC to Axin 1 and 2 for tumor suppression via destruction of β-catenin. These data present the uniquely important function of the Wnt/β-catenin pathway in the survival and proliferation of human cancers. The frequency and common nature of the mutations in this pathway during cancer development highlights the close connection of tissue patterning signals with oncogenicity.

Activation of the Wnt/β-catenin is widely regarded as oncogenic in nature, yet recent studies have shown that Wnt signaling also plays essential roles in tumor prevention. Elegant work by Brown and Pineda et al. showed that normal cells rely on Wnt secretion originating from β-catenin-driven neoplasms to control the physical removal of early aberrant growths [112]. These data show that Wnt signaling can play a critical role in triggering normal cells to remove pre-malignancies to maintain healthy tissue homeostasis throughout an animal’s lifespan.

## 11. New Approaches

### 11.1. Animal and Organoid Modeling

The study of self-organization is inherently challenging in vivo as it requires interrogating pathways at the level of tissue patterning across time in a complex microenvironment. To test feedback models, such as a RD model, monitoring the dynamics of morphogens from specific cells and their neighbors is required. To test an RD model fully, control of the morphogens and their spatial dynamics is important in differentiating between other potential mechanisms of gradient formation. Tuning the gradients predicted by a RD model and observing patterning at the tissue level over time is a task few have accomplished. Fortunately, multiple approaches exist to tackle these challenging problems. The combination of simple animal or mammalian organoid models with advances in live imaging, optogenetics, and microfluidic are likely to bring significant progress in revealing fundamental principles of biological self-organization.

Invertebrates biologists have been studying pattern formation during development for over half a century [113]. New technologies in *Drosophila* allow dynamic manipulation and monitoring of the Wnt pathway in animals by combining optogenetics and fluorescent reporter systems [114]. *Hydra* and *Planaria* are ideal models for the study of regeneration, as continuous stem cell regeneration takes place during normal animal growth, enabling impressive regeneration capabilities. These models can also be extensively manipulated genetically and are small and simple to maintain. Zebrafish patterning models can now be manipulated by optogenetics, allowing for noninvasive testing of a pioneering RD system [115].

In mammalian models, genetic systems have been used to study Wnt-driven follicle patterning in mice [37] using a lentiviral in utero delivery method [116]. Gradients of Wnt have been measured in the small intestine using live imaging of developing organoids and fluorescently tagged Wnt protein [63], allowing for mapping of Wnt gradients over time and under different genetic conditions. Two-photon microscopy of tissue-derived organoids combined with single cell-RNA sequencing has been a powerful tool in the intestine as well [18]. Using these tools, Serra et al. [18] uncovered a detailed YAP1-mediated mechanism for initiation of symmetry breaking in the intestine by tracking single cell phenotypes and gene signatures during organoid development. Our own group has developed a method for long-term culture of 2D organoids of the gut, and discovered a relationship between BMP and Wnt in maintaining the stem cell niche [29]. This system allows for high content screening and easily visualization of crypt patterning, which may elucidate new regulators of gut homeostasis and regeneration.

Microfluidics are another powerful tool to study self-organization. These devices allow for the manipulation of morphogens with high spatial precision via laminar flow or controlled diffusion [117]. These have been used to create “development-on-chip” approaches that can test the mechanisms of organizing centers such as neural tube patterning in a controlled contexed analogous to observed events in vivo [118]. The marriage of intestinal organoids with microfabrication and microfluidics has generated in vitro crypt-villus architecture that is amenable to signal gradient manipulation [119]. Future studies involving the precise control of morphogen concentrations will help define the necessity and sufficiency of key tissue organizing factors.

### 11.2. Computational Modeling

Building mathematical models is an underappreciated yet essential part of a productive understanding of tissue self-organization. Adequate understanding of mechanisms underlying biological self-organization is not even possible in principle without mathematical modeling. Take for example the case of intestinal crypts. Understanding the mechanism of their formation would not only require establishing the natural inter-cellular interactions, but also demonstrating that the established interactions are sufficient to reproduce the spatial pattern of the crypts. Clearly, the dynamics of a system this complex cannot be tracked mentally and thus requires computational/modeling approaches.

One of the earliest attempts at a computational description of Wnt pathway dynamics may be found in Reference [120]. There, using Xenopus egg extract as a model system, many key kinetic parameters of the pathway were estimated and incorporated into a computational model. The model correctly predicted the response of the pathway (measured as the rate of β-catenin depletion) to external biochemical/pharmacological perturbations. Subsequent analysis of the model, further experiments, and theoretical developments aimed to test model predictions are excellently summarized in Reference [121]. We posit that despite considerable progress and many significant findings, quantitative modeling of the Wnt pathway is still in its infancy. Specifically, it is by no means clear how comprehensively the key molecular players have been identified, how subcellular spatial compartmentation of pathway components influences pathway dynamics, or what/how cell–cell interactions control Wnt dynamics on the scale of the tissue. It appears to us that the key bottleneck in addressing these and related questions is the availability of quantitative data. A combination of multiple approaches will likely be instrumental in arriving at a more refined quantitative model of Wnt pathway dynamics. Firstly, in vitro reconstituted systems (e.g., Xenopus egg extract and organoids) are a major tool that enables high throughput/quantitative measurements. Such systems are straightforwardly combined with automated image processing, automated liquid handling, high throughput quantitative measurements and quantitative pharmacological/biochemical perturbations. Interpreting such data demands computational techniques to solve the inverse problem of mapping measurements to kinetic parameters describing the dynamics of pathway components. The most promising approaches will likely involve recent developments in Kalman filtering [122], Topological Data Analysis [123], and causal inference [124].

## 12. Conclusions

Answering many classical developmental and cell biological questions of the animal kingdom ultimately requires understanding the mechanisms that underlie the self-organizing potential of the Wnt pathway. Yet, many decades of research have failed to explain simple tissue forms such as the regular periodicity of intestinal crypts, or why damaged imaginal discs either regenerate or duplicate the organs of an adult fly depending on which part of a disc is ablated [125]. It now stands clear that answering these questions will require quantitative experimentation and modeling of Wnt pathway dynamics at the cellular and tissue level. Crucially, in this context, quantitation is not merely needed to dot the i’s and cross the t’s, but understanding qualitative outcomes of self-organization is achieved by describing the system quantitatively. Even though many major components of the Wnt pathway are known, the knowledge of those components alone is by no means enough to meaningfully interpret the phenotypes arising from genetic, pharmacological, or environmental perturbations. The use of reconstituted cellular systems together with quantitative modeling will pave the way.

## Figures and Tables

**Figure 1 genes-11-00939-f001:**
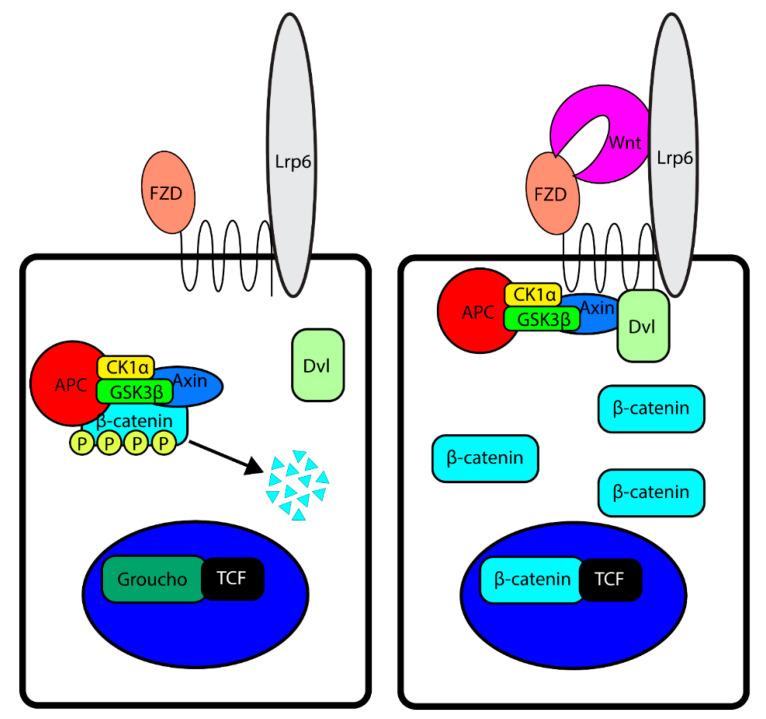
General model for the Wnt/β-catenin signaling pathway and its regulation by the destruction complex. The destruction complex: Axin, Adenomatous polyposis coli (APC), casein kinase 1 α (CK1α), glycogen synthase kinase 3 (GSK3), protein phosphatase 2A (PP2a), and the ubiquitin ligase βTrCP. Membrane receptors: Frizzled (FZD) and low-density lipoprotein receptor-related protein 6 (Lrp6). Dishevelled (Dvl) T-cell factor (TCF).

**Figure 2 genes-11-00939-f002:**
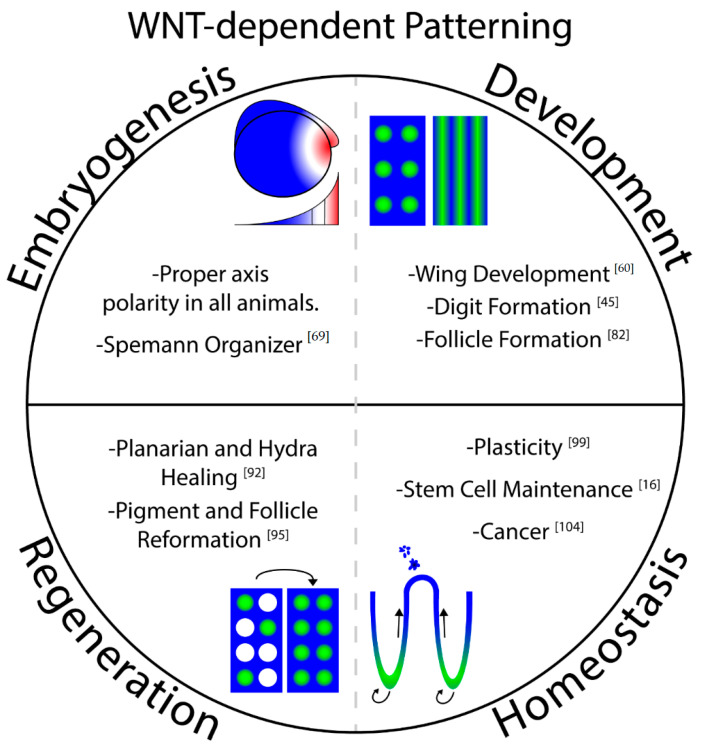
Wnt/β-catenin signaling in tissue self-organization. Two main categories are shown, Embryogenesis/Development (top) and Regeneration/Homeostasis (bottom). The categories were partitioned due to reversibility. Development of the embryo and body axis as well as appendage development are irreversible and temporally regulated patterning processes. Uniquely, regeneration and homeostasis are ongoing processes that an organism must call upon when needed. These processes have a greater need to be autonomous and could not withstand removal of a morphogen source unless there was a mechanism in place for sensing and replacing a missing niche.

**Figure 3 genes-11-00939-f003:**
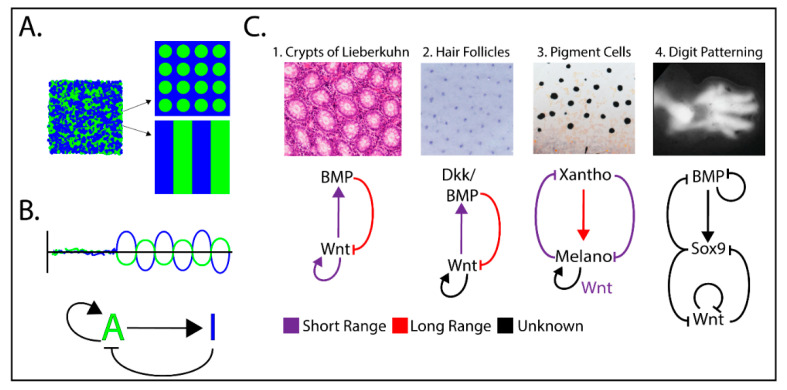
Examples of Wnt/β-catenin-dependent Reaction-Diffusion (RD) mechanisms implicated in tissue patterning. (**A**) Over time, a short-range activator and a long-range inhibitor interact to form local morphogen concentrations in patterns. (**B**) Basic model for the RD activator/inhibitor pair. (**C**) Examples of Wnt-driven patterning driven by RD mechanisms. Although the morphogen pairs/triplets and their mechanisms of control varies, the basic activator/inhibitor concepts are maintained and predicted by mathematical modeling. Patterning images C1–C4 were adapted from References [37,46,47,48].

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
