# Peer review of "Wnt/β-catenin Signaling in Tissue Self-Organization"

_genes, 2020, doi:10.3390/genes11080939_

Round 1

Reviewer 1 Report

The manuscript by Pond et al. is a valuable study by highlighting the history and literature on the role of Wnt proteins in pattern formation and on the “diffusion-reaction” (DR) morphogen models. While the Authors list many literature (and also own) data, our recent understanding reveals a more complicated scenery on both, the tissue self-organization and the canonical Wnt pathway than it is depicted in the manuscript.  

  • The canonical Wnt action stabilizes (prevents from ubiquitination) a number of proteins beside b-catenin. Depending on the type and differentiation state of the cells, it can elicit various cellular responses, among them several differentiation-promoting reactions (Herreros and Dunach 2019). Wnt proteins play varying roles in different stages of development and in different tissues, accordingly in the formation of different cell assemblies.
  • Some molecular relations (as an example, the inhibitory actions of BMPs on Wnt effects) can not be regarded as general phenomena.
  • I would discuss the universality of the statement “Embryogenesis and body development originates from an externally triggering one-way signalling cascade driven by external cues”. How could it fit for instance to the blastocyst formation?
  • Wnt proteins play important role in pattern formation, but the differential cell adhesion-based sorting mechanisms, those taking place as cell-autonomous motility mechanisms can not be disregarded if tissue organization is discussed.
  • In context with the formation of mammalian embryonic patterns and “germ layers” some morphogens are mentioned, while some of the most important players (as TGFb, SHH, etc) are not referred at all.
  • Several DR models are listed, but the manuscript does not provide a critical overview on their applicability to the recent understanding of Wnt actions

Author Response

Reviewer #1

The manuscript by Pond et al. is a valuable study by highlighting the history and literature on the role of Wnt proteins in pattern formation and on the “diffusion-reaction” (DR) morphogen models. While the Authors list many literature (and also own) data, our recent understanding reveals a more complicated scenery on both, the tissue self-organization and the canonical Wnt pathway than it is depicted in the manuscript.  

  • The canonical Wnt action stabilizes (prevents from ubiquitination) a number of proteins beside b-catenin. Depending on the type and differentiation state of the cells, it can elicit various cellular responses, among them several differentiation-promoting reactions (Herreros and Dunach 2019). Wnt proteins play varying roles in different stages of development and in different tissues, accordingly in the formation of different cell assemblies.

Thank you for this point, we agree and have added the referenced review you mentioned to the other B-catenin independent reviews listed in line 37

  • Some molecular relations (as an example, the inhibitory actions of BMPs on Wnt effects) cannot be regarded as general phenomena.

We have changed the wording regarding the Wnt/BMP relationship to highlight that their negative feedback may not always be the case, line 266. We also describe a pathway that is not negative feedback of BMP on Wnt in line 268

  • I would discuss the universality of the statement “Embryogenesis and body development originates from an externallytriggering one-way signalling cascade driven by external cues”. How could it fit for instance to the blastocyst formation?

We have modified this section to clarify our meaning here and added additional references. We realize that “external cues” may be a confusing term, so we removed it from our description. Please see lines 82-91

  • Wnt proteins play important role in pattern formation, but the differential cell adhesion-based sorting mechanisms, those taking place as cell-autonomous motility mechanisms can not be disregarded if tissue organization is discussed.

Great point! We have expanded on what we already mentioned about the mechanical forces at work during tissue organization by adding an additional section on this topic. We have added several sentences and 6 new references to highlight this important aspect of tissue development. Please see lines 76-80.

  • In context with the formation of mammalian embryonic patterns and “germ layers” some morphogens are mentioned, while some of the most important players (as TGFb, SHH, etc) are not referred at all.

We have added a text referencing the importance of other diffusible morphogens and 2 reviews covering some of their roles in embryonic tissue patterning, but because this review is Wnt focused, we did not have space to expand further on these other important players.  Please see lines 272-274.

  • Several DR models are listed, but the manuscript does not provide a critical overview on their applicability to the recent understanding of Wnt actions

Great point. We believe that RD models require much more investigation to explain Wnt-driven patterning. Currently, there are no examples of RD models which have been validated to explain Wnt-dependent tissue patterning. Our intent with this review is that combining math modeling with reconstitution biology and relevant tissue models described in our “new approaches” section will answer some of these questions.

Reviewer 2 Report

This manuscript provides an overview of the Wnt/β-catenin pathway in regards to its role in tissue patterning. It discusses various topics related to the role of Wnt during development and adult tissue homeostasis. The manuscript is very well written and very informative as it highlights historical milestones and new approaches such as computational modeling. Before the manuscript is published, however, the authors should consider addressing the following minor issues:

1) Although lines 38-53 essentially described Fig 1, Fig 1 should still contain some description. For ex., some of abbreviations such as FZD may be defined.

2) For Fig 2, cartoons and graphs should be described.

3) For Fig 3, break the figure it into sections. For ex., 3A, 3B, etc

Author Response

Reviewer #2

This manuscript provides an overview of the Wnt/β-catenin pathway in regards to its role in tissue patterning. It discusses various topics related to the role of Wnt during development and adult tissue homeostasis. The manuscript is very well written and very informative as it highlights historical milestones and new approaches such as computational modeling. Before the manuscript is published, however, the authors should consider addressing the following minor issues:

  • Although lines 38-53 essentially described Fig 1, Fig 1 should still contain some description. For ex., some of abbreviations such as FZD may be defined.

This change has been made. Lines 57-60

  • For Fig 2, cartoons and graphs should be described.

We attempted to describe the figure in clearer terms in lines 82-91

  • For Fig 3, break the figure it into sections. For ex., 3A, 3B, etc

This change has been made. See figure 3

Round 2

Reviewer 1 Report

I still would debate the bellow sentence:

line 91

“Embryogenesis and body development pathways are one-way signaling cascades which can be initiated by forces such as compartmentalized intracellular reactions [24] and controlled expression of maternally provided mRNA [25].”

  • I do not understand properly the “one-way signaling cascades” expression in relation to “embryogenesis and body development pathways”. All developmental processes are governed by multiple and competing factors including the distribution of counter-acting morphogens (as just one an example FGF8 and TGFb in initial anterior – posterior or BMP and SHH in lateral - medial axes formation in vertebrate embryonic plate), and the actual responsivity of the cells. In a given time-scale, all effects are reversible, while – it is true, that – we call them “development” only after they became irreversible.
  • Beside “compartmentalized intracellular reactions” (let us call them cell polarization) intercellular – as contact cell-to cell signalling (as Notch/Delta) and short-distance paracrine - interactions play inevitable roles in delineation of cell clusters and in initiating cell differentiation.
  • Maternally provided mRNAs are key-players in the early development of non-vertebrate embryos, but in the vertebrate embryos – those developing as multicellular entities from the very beginning – the morphogen-producing “organizers” (AVE, primitive streak, nodus) and their products play the orchestrating a roles.

Author Response

Thank you for bringing this point to our attention, we agree with you! We did not intend the sentence to be interpreted that way and have made changes to that paragraph to hopefully clear up the idea we meant to convey. Please see the paragraph beginning with line 81, which hopefully clears things up.